# Oxidative Stress, Inflammation and Connexin Hemichannels in Muscular Dystrophies

**DOI:** 10.3390/biomedicines10020507

**Published:** 2022-02-21

**Authors:** Arlek González-Jamett, Walter Vásquez, Gabriela Cifuentes-Riveros, Rafaela Martínez-Pando, Juan C. Sáez, Ana M. Cárdenas

**Affiliations:** 1Centro Interdisciplinario de Neurociencia de Valparaíso, Facultad de Ciencias, Universidad de Valparaíso, Valparaíso 2360102, Chile; wavasqueza@gmail.com (W.V.); juancarlos.saez@uv.cl (J.C.S.); 2Escuela de Química y Farmacia, Facultad de Farmacia, Universidad de Valparaíso, Valparaíso 2360102, Chile; gabriela.cifuentes@alumnos.uv.cl (G.C.-R.); rafaela.martinez@alumnos.uv.cl (R.M.-P.)

**Keywords:** muscular dystrophies, inflammation, oxidative stress, connexin hemichannels, resveratrol

## Abstract

Muscular dystrophies (MDs) are a heterogeneous group of congenital neuromuscular disorders whose clinical signs include myalgia, skeletal muscle weakness, hypotonia, and atrophy that leads to progressive muscle disability and loss of ambulation. MDs can also affect cardiac and respiratory muscles, impairing life-expectancy. MDs in clude Duchenne muscular dystrophy, Emery-Dreifuss muscular dystrophy, facioscapulohumeral muscular dystrophy and limb-girdle muscular dystrophy. These and other MDs are caused by mutations in genes that encode proteins responsible for the structure and function of skeletal muscles, such as components of the dystrophin-glycoprotein-complex that connect the sarcomeric-actin with the extracellular matrix, allowing contractile force transmission and providing stability during muscle contraction. Consequently, in dystrophic conditions in which such proteins are affected, muscle integrity is disrupted, leading to local inflammatory responses, oxidative stress, Ca^2+^-dyshomeostasis and muscle degeneration. In this scenario, dysregulation of connexin hemichannels seem to be an early disruptor of the homeostasis that further plays a relevant role in these processes. The interaction between all these elements constitutes a positive feedback loop that contributes to the worsening of the diseases. Thus, we discuss here the interplay between inflammation, oxidative stress and connexin hemichannels in the progression of MDs and their potential as therapeutic targets.

## 1. Introduction

Muscular dystrophies (MDs) are a heterogeneous group of diseases caused by mutations in genes encoding proteins with key functions for the muscle integrity. Despite their diverse genetic causes, MDs exhibit common clinical features that include progressive weakness and atrophy of specific muscular groups, e.g., distal limb, face, shoulder or upper arm and leg skeletal muscle, joint contractures, loss of ambulation, and respiratory and swallowing difficulties as the diseases progress [1]. Some MDs are also associated with cardiac disorders [2]. MDs’ aspects, such as their onset, severity, muscles affected, and rate of progression, mostly depend on the mutated gene. In this regard, mutations in around 50 genes have been associated with at least 70 different types of MD, which have been classified in nine different categories (see Table 1): Becker muscular dystrophy (BMD), congenital muscular dystrophy (CMD), distal muscular dystrophies (DiMD), Duchenne muscular dystrophy (DMD), Emery-Dreifuss muscular dystrophy (EDMD), facioscapulohumeral muscular dystrophy (FSHD), limb-girdle muscular dystrophy (LGMD), myotonic dystrophy (MiD), and occulopharyngeal muscular dystrophy (OMD). These MDs’ classifications are based on the clinical traits and age at onset, and they are subclassified according to their inheritance and the genetic bases of the disease [3,4]. Histopathologically, MDs are characterized by degeneration and necrosis of the muscle fibers, which are poorly regenerated and instead replaced by adipose and fibrotic tissue [5]. Another important trait of many MDs is chronic inflammation, as observed by the abundant presence of infiltrating inflammatory cells in dystrophic muscles. The persistent abundance of macrophages promotes the release of pro-fibrotic agents, including the transforming growth factor (TGF-β), that lead to excessive accumulation of extracellular matrix components, particularly collagen, and contribute to the formation of fibrotic tissue [5]. Another common feature in many MDs is the presence of oxidative stress (OS), which is characterized by the oxidation of lipids and proteins, and by the unbalance of the endogenous antioxidant systems [6]. Table 1 shows reported OS signs, inflammation markers and mitochondria traits, which are also involved in OS and inflammation as discussed later in different types of MDs.

The underlying pathomechanisms causing MDs are initiated by mutations in genes encoding proteins with dissimilar functions that, however, seem to converge in common cellular dysregulations resulting in OS and chronic inflammation. As we discuss here, MDs-causing mutations affect proteins critical to skeletal muscle integrity and homeostasis. Therefore, their defective forms or absence does not only cause particular cellular defects, but also activate signaling pathways that lead to OS and inflammation involving a positive feedback loop that contributes to the progression of MDs.

## 2. Signs of Inflammation in the Skeletal Muscle

Inflammation is a nonspecific mechanism driven by the immune system in response to harmful stimuli such as pathogen infection or damaged cells. Its purpose is to eliminate the cause of injury and to promote repair [41]. Inflammation occurs in vascularized tissue and is mediated by humoral and cell-factors that lead to leukocyte infiltration into the injured tissue. It has differentiated phases: acute, chronic, local and systemic. Acute inflammation is a highly regulated process that has a relatively short course and is solved once the noxious stimulus is removed [42]. Chronic inflammation is a long-lasting process, characterized by the proliferation of blood vessels, fibrosis and the simultaneous destruction and healing of the injured tissue [41]. Whereas local inflammation affects one organ or part of it, systemic inflammation affects the complete organism. In response to noxious stimuli, blood neutrophils and monocytes migrate towards the injured tissue. There, monocytes become macrophages that “engulf” the inflammatory-stimuli and release pro-inflammatory mediators promoting the recruitment of more immune-cells and amplifying the local acute inflammation [42]. At the peak of acute inflammation, pro-inflammatory molecules make a “transition” towards specialized pro-resolving mediators (SPMs) initiating the resolution of the local acute inflammatory response. Upon resolution, macrophages “switch” from a M1 pro-inflammatory to a M2 anti-inflammatory phenotype [42], facilitating this process. If this mechanism is dysregulated, pro-inflammatory signals persist and repair can be surpassed by damage, leading to chronic inflammation and tissue dysfunction that characterize the pathological conditions.

The skeletal muscle is a tissue permanently exposed to different traumas induced by contraction. Hence, it is constantly subjected to cycles of inflammation and repair. Skeletal muscles are organized in fascicles of between about 10 and 100 muscle fibers. These are long multinucleated cells whose nuclei are located in the periphery of the cell, not in the cell center [43]. Cylindrical organelles called myofibrils pack the contractile proteins myosin and actin. Each myofibril is organized into a variable number of sarcomeres, the functional unit for muscle contraction. The membrane of muscle fibers is called sarcolemma and presents deep invaginations named transverse tubules. Those are part of a network of endomembranes that surround myofibrils in association with the sarcoplasmic reticulum, the main intracellular store of Ca^2+^. One transverse tubule and two sarcoplasmic reticulum-cisternae constitute the “muscle triad”, the structural unit where the excitation-contraction coupling occurs [43]. Non-contractile elements provide structural support and facilitate the transmission of the force generated during contraction from sarcomere to all the muscle tissue. Among these elements are sarcolemmal-anchored proteins such as dystrophin; transmembrane proteins such as β-dystroglycan, sarcoglycan and integrins; and extracellular matrix-proteins such as the α-dystroglycan and laminin-2 [44].

Under physiological conditions, most skeletal muscles are adaptable tissues that can undergo structural and functional modifications in response to different stimuli (i.e., use, hormonal and nutrient status, a process called muscle plasticity) [45]. In addition, and in response to intense exercise or injury, they are able to fully regenerate, recovering the number and/or the size of muscle-fibers [46]. The nuclei of muscle fibers do not undergo divisions; thus, under conditions in which muscle tissue must be repaired, the response is performed by muscle-stem cells called “satellite cells”. These cells express the myogenic transcription factor Pax7 [47] and are located in a specialized region between the sarcolemma and the basal lamina in a state of “quiescence”. After muscle injury, satellite cells are activated, becoming MyoD+ and myogenin+ myoblasts that undergo fusion and form syncytia, and upon innervation, they differentiate into mature fibers [47]. In addition to the activation of satellite cells, a time-dependent local acute inflammation is triggered in skeletal muscles after injury (Figure 1(A1)). This process is critical for the recruitment of immune cells that contribute to muscle regeneration [48]. At the onset of the muscle injury, the complement system is activated, promoting inflammatory cascades that lead to the infiltration of monocytes, neutrophils and T-cells [49,50]. Monocytes acquire a M1-proinflammatory profile that stimulates the proliferation of satellite-cells by releasing growth factors and proinflammatory cytokines (Figure 1(A1)) [51]. Among them tumor necrosis factor alpha (TNFα), interleukin (IL)-6 and IL-15 act as promoters of the myoblast fusion and myotube formation [52,53]. These and other proinflammatory cytokines are also secreted by T-cells [54] importantly contributing to the process. Thereafter, a resolution phase proceeds in which M1 macrophages switch to M2 further promoting differentiation of myoblasts into myotubes, growth of muscle fibers (Figure 1(A2)) and skeletal muscle regeneration (Figure 1(A3)) [55,56,57]. In healthy conditions, these events resolve rapidly once the cause of injury has been “removed”. However, in pathological contexts in which the stressor-agent persists the M1 to M2 transition is impaired becoming skeletal muscle inflammation chronic (Figure 1B). The latter might produce accumulation of fibrotic and fat tissue (Figure 1B), making regeneration inefficient and contributing to the pathophysiology of muscle diseases [58]. Persistent immune cells as well as muscle fibers release an exacerbated amount of proinflammatory cytokines enhancing the activation of signaling mediated by the nuclear factor kappa B (NF-kB), a key inductor of the transcription of pro-inflammatory genes (Figure 1B). Parallel the assembly of the NLRP3 inflammasome leads to the activation of the caspase-1 enzyme and the subsequent proteolysis of pro-IL-1β and pro-IL-18 (Figure 1B) further promoting inflammation and perpetuating the inflammation/regeneration cycle (Figure 1B).

In MDs, mutated proteins critically affect the homeostasis of skeletal muscle, causing damage to the sarcolemma and contractile apparatus and leading to the persistence of the inflammatory events. The accumulation of proinflammatory mediators in the skeletal muscle tissue affects its regenerative capacity, impairing the healing mechanisms [58]. Consequently, muscle is replaced by fibrotic tissue and fat [5], leading to muscle dysfunction. Hence, chronic inflammation has been described in the context of different MDs. In the most severe forms of MDs, it progresses, affecting other muscles and tissues becoming a systemic chronic inflammation, which appears to contribute to the progressive muscle weakness and atrophy. Among MDs that reportedly develop with chronic inflammation are DMD [59], LGMD [35], merosin/laminin-deficient congenital muscular dystrophy [60], Miyoshi myopathy (MM) [15,16] and LMNA-related myopathies [61]. Chronic inflammation also appears to contribute to the atrophy in non-dystrophic myopathies such as in the valosin-containing-protein myopathy [62] and in the sporadic late-onset nemaline-myopathy [63].

## 3. Chronic Inflammation in Skeletal Muscle Dystrophies

DMD is the most frequent MD with an estimated worldwide prevalence of 4.78 per 100,000 births [64]. It is caused by X-linked mutations in the DMD gene encoding dystrophin. This protein is believed to be a component of the dystrophin glycoprotein complex (DGC) that connects the actin cytoskeleton with the extracellular matrix providing stability and structural integrity to the sarcolemma during muscle contraction [65]. DMD mutations produce dystrophin deficiency, leading to membrane depolarization, fragility, and high susceptibility to injury during muscle contraction [66]. Such defects associate to massive immune cell infiltration, chronic inflammation, and necroptosis, a programmed form of necrosis [67]. In this regard, mdx mice, a mammalian model of DMD, display an enhanced susceptibility to sarcolemma rupture under mechanical stress [68]. Upon DMD, skeletal muscles lose their ability to regenerate, resulting in fibrosis, early onset muscle dysfunction, and eventually loss of ambulation [20,69,70]. One variant of this disease is BMD. It is also caused by X-linked mutations in the dystrophin gene, but in this case, they preserve the reading-frame, allowing the expression of a partially functional truncated form of dystrophin [71,72]. Such differences result in a milder form of dystrophy in BMD, with a slower progression than that observed in DMD [72].

Local and systemic inflammation have been reported to be associated with the fibrosis and atrophy in DMD. Overexpression of pro-inflammatory cytokines such as IL-6, IL-1β and TNF-α [20], as well as anti-inflammatory cytokines such as IL-10 and TGF-β [73], are chronically detected in DMD. The over-activation of the NF-kB (Figure 1B), has also been reported in DMD patients and animal models [69]. Congruently, the inhibition of NF-κB has been postulated as a potential therapeutic target in Duchenne. In 2016, Hammers and collaborators used the NF-κB inhibitors edasalonexent and CAT-1041 to treat the dystrophy in mdx mice. The oral administration of these NF-κB inhibitors, which were composed by omega-3- polyunsaturated fatty acids (PUFAS) conjugated to salicylic acid, improved the dystrophic phenotype in terms of activity and muscle mass, reduced inflammation and fibrosis [74]. In the same line, treatment with eicosapentaenoic acid (EPA) has further shown to protect against the muscle damage in the mdx mice by promoting a shift from the M1 to M2 macrophage phenotype [75]. In placebo-controlled, double-blind, randomized trials carried out in DMD patients, treatment with EPA and docoshexaenoic acid (DHA) reduced inflammation markers and diminished the expression of NF-κB in leukocytes [76,77]. Furthermore, in phase 2 and 3 clinical trials in DMD young patients, NF-κB inhibitors such as flavocoxid or edasalonexent, showed to reduce the serum levels of IL-1β and TNF-α, slowed-down the disease progression and preserved muscle function [78,79,80], indicating that NF-κB might be a promising targeted therapy for MDs.

Another important mediator of the inflammatory process that is over-active in neuromuscular diseases including DMD is the NLRP3 inflammasome [81]. Inflammasomes are cytosolic-receptors of the innate immune system that assemble in response to harmful stimuli, mediating the activation of caspase-1 [82] (Figure 1B). This enzyme catalyzes the proteolytic processing of inactive precursors of IL-1β and IL-18, turning them into their active forms [82] and promoting inflammation (Figure 1B). The NLRP3 inflammasome is also expressed in skeletal myofibers [35] and its inhibition has been evaluated as a potential therapeutic target in DMD. In 2018, Boursereau and collaborators demonstrated that Adiponectin (ApN), an adipocyte-secreted cytokine that regulates glucose and fatty-acid metabolism [83], exerts downregulation of NLRP3 via the micro-RNA miR-711 [59]. Congruently, overexpression of ApN seems to protect skeletal muscle against inflammation and injury, as well to improve the muscle function in mdx mice [84]. The positive regulation of the ApN receptor reduced infiltration of T-cells and promoted the transition from M1 to M2 macrophages in the muscles of mdx mice, promoting regeneration [85]. More recently, AdipoRon, an ApN-receptor agonist, has been used in mdx mice with similar results. The oral administration of AdipoRon for eight weeks managed to protect the mdx skeletal muscle against chronic inflammation and OS, attenuating the dystrophic phenotype [86]. In the same line ghrelin, another metabolic hormone that participates in the regulation of the appetite, exerts a similarly positive effect and has been suggested as a potential candidate for the DMD treatment. Ghrelin possesses anti-inflammatory activity, prevents skeletal muscle atrophy, increases muscle regeneration, and improves the dystrophic phenotype rescuing the muscle function in *mdx* mice [87]. As with ApN, the ghrelin action seems to rely on the inhibition of the NLRP3 inflammasome assembly [87].

In addition to the evidence reported in the DMD context, inflammation has been also described as part of the pathological mechanisms in other MDs. Merosin/laminin- congenital muscular dystrophies, EDMD, and LGMD constitute other examples of inflammatory diseases. The merosin-deficient congenital muscular dystrophy type 1A (MDC1A) is caused by mutations in the LAMA2 gene [60] that lead to the partial or complete absence of α2-laminin. This protein, also called merosin, is a component of the extracellular matrix (ECM) that links the ECM to the DGC and to the sarcolemma-associated integrin complex [46]. Consequently, merosin plays a critical role in the maintenance of the sarcolemma integrity and muscle function. Mutations in other genes that code components of the ECM, such as the collagen type VI, are associated with other MDs, including Bethlem myopathy and Ullrich-scleroatonic muscular dystrophy [88]. The expression of merosine in satellite cells is associated with the proliferation and differentiation of myogenic cells [46]. Consequently, MDC1A-causing mutations in the LAMA2 gene lead to a defective muscle repair associated with chronic inflammation, fibrosis, and muscle atrophy [89]. Clinical symptoms of MDC1A include severe muscle atrophy, progressive muscle weakness, joint contractures, breathing and feeding difficulties [60]. Most patients lose their ambulation in infancy and exhibit a drastic shortening of their lifespan [89]. In skeletal muscle of patients and mouse models of MDC1A, an early onset of chronic inflammation occurs, leading to cycles of degeneration/regeneration and accumulation of fibrotic lesions [60]. In DyW mice, a murine model of laminin-deficient muscular dystrophy, merosin-deficient skeletal muscles exhibit high macrophage infiltration from early ages to adulthood. This infiltration is accompanied by an increase in the NF-κB signaling and over-expression of pro-inflammatory cytokines, which favors inflammation and causes inhibition of myogenesis [89].

EDMD is an early onset dystrophy characterized by slowly progressive muscle atrophy and weakness, spinal stiffness and heart disease. The overall prevalence of EDMD is unknown, but it is estimated to be the third most prevalent muscular dystrophy with an estimated of 1 case per 100,000 newborns [90]. EDMD is mostly caused by X-linked mutations in the EMD gene encoding emerin and by autosomal mutations in the LMNA-gene encoding lamin A/C [90]. Mutations in these genes have a negative impact on several functions of the nuclear envelope, such as the nuclear structure, cell signaling, and gene expression [61]. Mutations in lamin A/C and laminin-binding proteins are associated with the activation of the transcription factor NF-κB and with the consequent secretion of proinflammatory cytokines. A 2018 study carried out by Cappelletti et al. [61] demonstrated that myoblasts from patients with mutations in the LMNA gene secrete high amounts of pro-inflammatory cytokines, such as IL-6 and IL-8, in a similar way to what occurs with factors secreted by senescent cells or persistently damaged cells. In fact, mutations in the LMNA gene cause structural lesions of the nuclear lamina in macrophages, inducing a modification of their adhesive properties and promoting their infiltration in skeletal muscles [61].

Mutations in sarcoglycan components of the DGC lead to LGMD [91]. It is a heterogeneous group of MDs that primarily affects shoulders and hips. Its clinical manifestations range from severe forms with neonatal-onset to milder late-onset and slowly progressive forms. Mutations in genes encoding the caveolae-associated protein caveolin-3 [92], the Ca^2+^-regulated proteolytic enzyme calpain-3 [93], the component of intermediate filaments desmin [94], the giant sarcomeric protein titin [95] or the glycosyltransferase-enzymes are also associated with different forms of LGMD [3]. Recessive mutations in the DYSF gene, which encodes the protein dysferlin, cause LGMD type 2B (LGMD2B) and MM [96]. Dysferlin is expressed in skeletal and cardiac muscle cells, as well as in monocytes and macrophages playing a key role in membrane fusion and repair [97]. It interacts with annexin-A1 and annexin-A2, which are Ca^2+^-binding molecules involved in the sarcolemma-repair [97]. The membrane-repairing function in dysferlin-deficient myofibers can be recovered in dysferlinopathy models by expressing a “mini-dysferlin” peptide or myoferlin, another ferlin family protein, but these approaches do not arrest muscular degeneration [98]. In this regard, an additional function of dysferlin has been proposed, which could be implicated in the pathological mechanisms of dysferlinopathy. As dysferlin has been localized in intracellular vesicles [99], regulates the cytoskeletal actin remodeling [100] and has been detected in non-mechanically active tissues including endothelial cells, where its absence causes deficient trafficking of membrane-bound proteins [101], it is likely that dysferlin acts as a mediator in the traffick of other proteins. Thus, the main dysferlin dysfunction caused by dysferlinopathy-linked mutations requires further study. In this regard, the mechanisms contributing to the onset and progression of this type of LGMD are not yet fully defined, although mutations in the DYSF gene are associated with the loss of dysferlin expression and seem to compromise the capability of myofiber for repair following sarcolemmal injury. The latter leads to chronic inflammation, degeneration, and gradual adipogenic replacement of the muscle tissue [97]. This has been demonstrated through the involvement of immune factors in the pathogenesis of dysferlinopathy [102]. In fact, the absence of dysferlin favors intramuscle macrophage recruitment, proliferation, and skews macrophages toward a cyto-destructive phenotype [103]. As activated macrophages are a rich source of radical species and proinflammatory cytokines, their infiltration in muscles exacerbates local damage. In fact, several studies suggest that these mediators make dysferlin-deficient muscles more vulnerable to damage. Remarkably, the suppression of the innate-immune response mediated by toll-like receptors (TLRs) has been shown to reduce the atrophy and improve muscle strength in a dysferlin-deficient mouse model [104]. These data are consistent with the research carried out by Rawat et al. in 2010 [35], who showed that TLR-mediated signaling pathways trigger an inflammatory response that involves the activation of NF-κB and the assembly of the NLRP3 inflammasome in the dysferin-deficient dystrophic muscles [29,35].

Acute inflammation is necessary to trigger a repair-program that regenerate muscles after injury [48]. However, the mechanisms that transform this process in a chronic condition contributing to muscle degeneration remain poorly understood. A possible explanation could be related to the large amount of reactive oxygen species (ROS) produced by the inflammatory cells, such as macrophages and neutrophils, which infiltrate the skeletal muscle damaged [41]. As chronic inflammation seems to be importantly involved in the progression of MDs, this is an aspect to be considered in the search of new potential therapeutic targets.

## 4. Oxidative Stress in Muscular Dystrophies

In biology, ROS are endogenous molecules produced by different tissues under physiological conditions, although they can be overproduced in pathological states. In the skeletal muscle, ROS such as the radical superoxide (O_2_^•−^) and its dismutation product hydrogen peroxide (H_2_O_2_) are transiently produced during high metabolic demand and ATP depletion, for instance during physical activity [105]. Major sources of ROS include mitochondria [106], purine metabolism by xanthine oxidase [107], oxygenases that metabolize arachidonic acid (lipoxygenases and cyclooxygenases) [108] and the nicotinamide adenine dinucleotide phosphate (NADPH) oxidase complex (Figure 2) [109]. The main source of O_2_^•−^ during exercise comes from the activation of the NADPH-oxidase (NOX) complex [110]. This membrane-bound enzyme system catalyzes the O_2_^•−^ production by transferring an electron from NADPH to the diatomic oxygen (O_2_). During exercise, the NOX system seem to be stimulated by either membrane depolarization or protein kinase C (PKC) activation [111,112]. PKC activation results from a positive feedback loop generated by the release of ATP through pannexin channels with the consequent activation of purinergic P2Y1 receptors [112,113]. Increased levels of NADH during contraction can also contribute to O_2_^•−^ production via the NOX system [110]. In addition, activation of the insulin receptor in skeletal muscle cells constitutes an additional mechanism for ROS production, as it promotes the activation of NOX, through a mechanism that involves phosphoinositide 3-kinase (PIK3) and PKC [114]. In turn, ROS production during physical activity favors Ca^2+^ release through the ryanodine receptor [115] and contributes to about 50% of glucose uptake through the glucose transporter 4 (GLUT4) [116]. This mechanism requires the translocation of GLUT4 from intracellular stores towards the sarcolemma and transverse tubules [117]. The contraction-induced GLUT4 translocation is impaired when muscles are pre-incubated with ROS-scavengers [118], suggesting that ROS are necessary for contraction-mediated glucose uptake. Upon moderate exercise, the small GTPase Rac1 is activated, promoting actin polymerization and consequently favoring the actin-mediated GLUT4-translocation [119]. In this regard, mice deficient in Rac1 or the NOX2 subunit p47phox exhibit a reduced ROS production and impaired glucose uptake in response to moderate exercise [116]. It has also been reported that ROS participate in exercise-induced mitochondrial adaptations [120], via a mechanism that involves the nuclear factor erythroid-derived 2-related factor 2 (Nrf2) [121]. Major mitochondrial changes produced by endurance-training in response to exercise are modifications in their content, biogenesis, fusion and segmentation [122].

Skeletal muscles also produce reactive nitrogen species (RNS) such as nitric oxide (NO^•^), peroxynitrite (ONOO^−^), nitroxyl (HNO) and nitrosonium cation (NO^+^), among others [123]. NO^•^ is a highly diffusible molecule with a very short biologic half-life (0.1–2 s) that under physiological conditions acts as a second messenger by stimulating the guanylyl cyclase (GC) and cyclic guanosine monophosphate (cGMP) pathways. During contraction, NO^•^ is generated by the activation of the splice variant μ of the neuronal nitric oxide synthase (nNOS), via a mechanism that seems to be mediated by the nNOSμ phosphorylation by AMPK [124]. In turn, the NO^•^ produced during contraction reportedly acts as a paracrine vasodilator that counteracts the sympathetic vasoconstriction produced during exercise [125]. It has been also proposed that NO^•^ promotes glucose uptake [124]. However, glucose uptake in gastrocnemius muscle during exercise is not impaired in nNOSμ-knock-out (KO) mice [126]. It is therefore possible that other NO^•^ sources or mechanisms contribute to the exercise-induced glucose uptake.

Although moderate levels of ROS are necessary for normal contraction and force production, the excess of ROS can lead to muscle fatigue and contractile dysfunction [105]. Under physiological conditions different enzymes, such as superoxide dismutase (SOD), catalase, glutathione peroxidase and thioredoxin reductase, and antioxidant endogenous molecules such as glutathione (GSH), prevent ROS accumulation [105]. Either catalase or glutathione peroxidase reduce H_2_O_2_ to H_2_O. On the other hand, RNS can be neutralized by hemoglobin, uric acid, β-carotene, vitamins E and C, as well as by SOD, glutathione peroxidase and thioredoxin [127]. These antioxidant systems allow a fine-tune balance of ROS levels, and disruption of this equilibrium can cause OS, a condition implicated in the progression of different skeletal muscle affections, including sarcopenia and different types of myopathies [6,128]. The redox imbalance may be due to an increased production of O_2_^•−^ or a reduced neutralization of ROS and RNS. Thus, persistent high levels of O_2_^•−^ and H_2_O_2_ may release metal ions, such as copper and iron, from their respective protein complexes to favor the production of hydroxyl radical (HO^•^) by a Fenton-type reaction mechanism [129]. HO^•^ is one of the most reactive ROS that has an oxidative potential of 1.9 V at pH 7.0 and a half-life of 10^−9^ s, properties that mean that its harmful actions may be limited to a restricted area [130]. As O_2_^•−^ can react with NO^•^ to generate ONOO^−^, another strong oxidant that further reacts with carbon dioxide (CO_2_) to form the radicals CO_3_^•−^ and NO_2_^•−^, or the adduct ONOOCO_2_^−^ [127], high levels of ROS can also trigger nitrosative stress. These different ROS and RNS can produce post-translational modifications such as oxidation, S-nitrosylation, S-glutathionylation or S-tyrosine-nitration of proteins, lipid peroxidation, and oxidation and nitration of nucleic acids (Figure 2), thus causing disfunction of different cellular elements and processes and deleterious effects to cells [127,131]. Furthermore, high ROS generation can also activate cellular signaling pathways that lead to apoptosis or necrosis, such as the c-Jun N-terminal kinases (JNK)/p53 pathway [132] and receptor-interacting serine/threonine protein kinase (RIP) 1 and 3 complexes [133], respectively. Among the target proteins susceptible to modification by ROS is NF-κB [134]. As aforementioned, this signaling is involved in different skeletal muscle physiological processes, but, its chronic activation has been observed in different skeletal muscle disorders, including MDs [29,89,135]. ROS can both directly modify NF-κB and its upstream kinases, and these can lead to either activation or repression of the NF-κB signaling (Figure 2); in turn, NF-κB activation can promote either anti- or pro-oxidant responses [136]. In this case, the interplay between ROS and NF-κB depends on the context and stage of the muscle redox state. ROS can also influence the activity of the Nrf2 (Figure 2). Under OS, this nuclear factor translocates to the nucleus to promote the expression of many antioxidant enzymes that render protection upon a stronger OS-promoting condition [137]. In contrast, its deficiency enhances ROS-induced damage in dystrophic muscles [31], whereas its activation seems to mitigate the progress of these diseases [69], presumably acting as a preconditioning factor.

A critical interplay during OS is which occurs between ROS and mitochondria, the main source for ATP production in skeletal muscles during aerobic respiration. These organelles also produce ROS as a result of the electron leakage at the electron transport chain during basal respiration [138]. The O_2_^•−^ generated, less than 1% of the total fraction of O_2_ utilized [139], is neutralized by the Mn-SOD (SOD2) and Cu, Zn-SOD (SOD1) present in the mitochondrial matrix and intermembrane space, respectively [140]. However, ROS production by mitochondria drastically increases under danger signals, as observed during MDs (Figure 2), wherein proinflammatory cytokines, or high levels of cytosolic Ca^2+^ and Na^+^ that reduced the mitochondrial membrane potential promote mitochondrial ROS generation [141]. During physiological Ca^2+^ signals, mitochondria uptakes Ca^2+^ from the cytosol essentially through a uniporter Ca^2+^ channel in the inner mitochondrial membrane [142]. This Ca^2+^ uptake stimulates the oxidative metabolism in the mitochondrial matrix by regulating Kreb cycle enzymes, such as the isocitrate-dehydrogenase, α-ketoglutarate dehydrogenase and pyruvate dehydrogenase [143]. However, excessively high cytosolic Ca^2+^ concentrations overcharge the mitochondrial Ca^2+^, inducing the constant opening of the mitochondrial permeability transition pore, a non-specific pore that allows the passage of molecules of <1.5 kDa to the mitochondria matrix [144]. A high and persistent conductance of this pore can lead to osmotic swelling, rupture of the outer membrane, and metabolic collapse [145]. Furthermore, OS can also induce an imbalance in mitochondrial fission-fusion, promoting their fragmentation [146], thus contributing with an additional mechanism inducing cell damage.

OS has been found in different types of MD, being more extensively studied in DMD. Indeed, OS signs, such as nucleotide oxidative products, oxidized glutathione or lipid peroxidation products, have been found in patients with DMD [17,18,19] and animal models of the disease [147,148]. An oxidative imbalance has also been reported in other MDs such as the LGMD2B and MM caused by dysferlin mutations. In this regard, muscle biopsies of patients suffering from dysferlinopathy show increased levels of protein oxidation and lipid peroxidation and altered reduced glutathione and antioxidant enzyme activity [27,28,29]. Dysferlin-deficient A/J mice also show high levels of ROS in *flexor digitoris brevis* (FDB) myofibers [30,31] and protein thiol oxidation in quadriceps [32]. As we recently reported, quadriceps and gastrocnemius muscles of dysferlin-deficient Bla/J mice, another model of dysferlinopathy, exhibit high levels of protein oxidation and lipid peroxidation and altered activity of the antioxidant enzymes superoxide dismutase and catalase [33]. In this regard, the treatment of dysferlin-deficient Bla/J mice with the antioxidant agent N-acetylcysteine reduces the OS signs and appears to improve muscle strength and/or resistance to fatigue [33]. A common feature that might induce OS in MDs is the absence or malfunction of sarcolemma integrity-associated proteins such as dystrophin and dysferlin, which can lead to an altered Ca^2+^ homeostasis (Figure 2) [96,149,150]. The mechanism by which these types of MDs show altered Ca^2+^ levels is not clear, and it has been proposed that it is a consequence of a non-specific Ca^2+^ influx through microtears in the sarcolemma (Figure 2) [149,150]. However, as we discuss later, the de novo expression of connexin (Cx) hemichannels and overexpression of pannexin (Panx) might further contribute to the Ca^2+^ dyshomeostasis in MDs such as DMD and dysferlinopathy (Figure 2) [151,152]. Importantly, the interplay between the Ca^2+^ homeostasis and ROS is bidirectional. Indeed, persistent high intracellular Ca^2+^ levels can contribute to mitochondrial Ca^2+^ overload, NOX-activation and ROS overproduction (Figure 2) [153,154]. In turn, ROS production by NOX4 contributes to the S-nitrosylation of the ryanodine receptor 1 (RyR1), increasing the Ca^2+^ leak from the sarcoplasmic reticulum (Figure 2) [154].

An altered oxidant status has also been found in EDMD [24,25]. This skeletal muscle disorder is caused by mutations in the gene encoding lamin A/C that increase the basal levels of ROS and lead to mitochondrial dysfunction [155]. It has been hypothesized that conserved cysteine residues in the lamin A C-terminal neutralize ROS, thus preventing protein oxidation [156]. It has also been proposed that lamin-A mutations associated to EDMD induce an atypical nuclear localization of the stress responsive protein Ankrd2, increasing the susceptibility to ROS damage [157]. On the other hand, muscle-specific expression of the mutant lamins in Drosophila produced reductive stress caused by cytoplasmic protein aggregation [158].

An increased susceptibility to ROS has also been observed in FSHD [159]. OS signs such as high levels of lipid peroxidation, protein carbonylation and DNA oxidation, and mitochondrial dysfunction have been found in skeletal muscles biopsies and blood samples from FSHD patients [26]. This muscle affection was initially associated with hypomethylation of a region known as D4Z4 in the chromosome 4q35, which contains the double homeobox 4 (DUX4) gene. However, recent analyses show that the D4Z4 hypomethylation did not correlate with the disease status [160,161] and, instead, it seems to be due to the chromatin structure present in the contracted allele [161]. Immortalized human myoblasts expressing DUX4 exhibit high levels of ROS [162], and in turn, OS increases the DUX4 expression [163]. In this regard, a transcriptome analysis in DUX4-expressing myoblasts identified 200 genes relevant for the FSHD pathogenesis that are deregulated by DUX4 indirectly, through OS [162]. Interestingly, silencing one of such genes managed to restore the differentiation ability of DUX4-expressing myoblasts [162].

## 5. Connexin and Pannexin Channels in Muscular Dystrophies

The Cx gene family comprises 20 members in mice and 21 genes in the human genome, of which 19 can be grouped as orthologous pairs [164]. These proteins form poorly selective membrane channels called hemichannels, which are present in the cell membrane of most cell types allowing communication between the cytoplasm and the extracellular medium. In addition, Cxs can form gap junction channels that communicate the cytoplasm of cells in contact [165]. Cx hemichannels also participate in autocrine and paracrine communication, as they are permeable to different small metabolites and molecules involved in cell signaling, such as glucose, NAD+, NO, prostaglandin E2, glutamate, and ATP [166]. In addition, some of them (i.e., Cx26 and Cx43) have been shown to be permeable to Ca^2+^ [167,168], whereas others (i.e., Cx39) might be rather impermeable to this divalent cation [169].

Other poorly selective channels are those constituted of Panx, a family of glycoproteins composed of 3 members, Panx1-3 [170]. Panx1 is ubiquitously expressed in various tissues, while Panx2 is primarily expressed in the central nervous system [171]. Panx3 is found in bone, skin, and skeletal muscles [170,172]. Panx1 channels localize in the cell membrane and play a relevant role in the release of ATP from different cell types [170]. Different stimuli capable of activating Panx1 channels have been reported, including voltage [173], membrane stretching [174,175], intracellular Ca^2+^ augmentation [175,176], and its C-terminal domain proteolysis [177]. However, it was recently shown that Panx1 is not directly sensitive to stretching instead, stretch stimulates other channels that allow the entry of Ca^2+^ promoting the activation of the Ca^2+^/calmodulin-dependent protein kinase II (CaMK ll), which phosphorylates Panx1, causing a conformational change of the channel that allows the passage of ATP through its “lateral tunnels” [175].

In skeletal muscles, myoblasts express Cx-based hemichannels and gap junction channels essential during myogenesis [178]. However, at terminal stages of the myogenesis, the expression of Cxs is inhibited [178] in a manner dependent on the release of acetylcholine by motoneurons [179]. In the adult stage, skeletal muscles present a preform Cx mRNA [179], suggesting that they might be rapidly translated under particular conditions which are not yet identified. Indeed, the expression of Cxs 39, 43 and 45 occurs transiently during regeneration after damage [180,181]. On the other hand, the expression of Panx1 is very low in undifferentiated myoblasts but drastically increases upon muscle differentiation, playing a key role in this process [182,183].

The de novo expression of Cx-hemichannels has been implicated in skeletal muscle inflammation by acting as a pathway for the release of ATP which is the source of extracellular adenosine that mediate preconditioning [184]. In addition, preconditioning has been associated to the translocation of Cx43 to the mitochondria where it can affect the distribution of K^+^ across the inner membrane [185]. On the other hand, metabolic inhibition increases plasma membrane expression, nitrosylation and opening of Cx43 hemichannels [186] and nitrosylation at the residue C271 promotes the opening of Cx43 hemichannel, as observed in mdx mouse-cardiomyocytes [187]. A crosstalk between Cx hemichannels and OS has also been demonstrated, as inhibition of Cx hemichannels protects cells from OS [188,189,190] and Cx hemichannels are inhibited by antioxidant agents such as resveratrol and α-tocopherol and partially nhibited by N-acetylcysteine [191]. Therefore, in addition to the intrinsic antioxidant effects of some compounds, their protective effect could include prevention of the rise in intracellular Ca^2+^ signaling known to activate several metabolic pathways that generate superoxide anion as described above. Remarkably, myofibers deficient in Cx43 and Cx45 expression do not present mitochondrial dysfunction induced by a synthetic glucocorticoid, suggesting that mitochondrial impairment occurs downstream of de novo Cx hemichannel expression, which reduces the membrane potential and consequently reduces the ionic asymmetry across the sarcolemma, affecting normal mitochondrial functioning [191].

In the pathological context of MDs, Cx and Panx hemichannels have been implicated. Panx1 over-expresses, together with the de novo expression of Cx hemichannels, contributing to increase the permeability of the sarcolemma [151,152]. In mdx mice, in which the absence of dystrophin is associated with an increase in the intracellular Ca^2+^ [149,150] that lead to myofiber necrosis [192], the de novo expression of Cx39, Cx43 and Cx45 has been demonstrated [151]. Interestingly, mdx mice in which a myofiber-specific KO of Cx43 and Cx45 was performed, presented normal intracellular Ca^2+^ signals, absence of myofiber apoptosis, reduced myofiber necrosis and improved muscle function [151]. The latter strongly suggests that a Cx hemichanel-mediated Ca^2+^ dyshomeostasis could importantly contribute to the pathophysiology of MDs (Figure 2). The role of elevated cytoplasmic Ca^2+^ in cell death is supported by experiments in which the overexpression of the transient receptor potential canonical 3 (TRPC3) in myofibers leads to increased Ca^2+^ influx, resulting in a dystrophic phenotype [192]. On the contrary, the transgenic suppression of TRPC channels in mdx mice reduced the Ca^2+^ influx and dystrophic signs [192]. However, in these studies the sarcolemma’s permeability to Evans blue is not observed, and rather is completely abrogated in myofibers deficient in Cx43 and Cx45 [151,152], indicating that Cx hemichannels and not sarcolemma microtears are the initial step of myofiber damage in MDs.

Panx1 channel expression is also enhanced in mdx mice [151,172,193]. It is accompanied by an increase in the Panx1-mediated ATP release, which promotes the increase in the intracellular Ca^2+^. In this respect, it was shown that the treatment with the L-type voltage-dependent-Ca^2+^-channel inhibitor nifedipine reduced the extracellular ATP levels in mdx myofibers by reducing the intracellular Ca^2+^ concentration [193]. The nifedipine treatment also reduced the expression of the NOX2 system, which is overexpressed in the diaphragm of mdx mice where it associates to the overproduction of ROS and decreased respiratory function [193]. At the level of cardiac muscle, there is an aberrant expression of Cx43 hemichannels in mdx mice, which is associated with an altered Ca^2+^ signaling. Congruently, a reduction in the Cx43 hemichannel activity seems to improve Ca^2+^ signaling, as well as to reduce NOX2 and ROS production, protecting mdx mice from inducible arrhythmias and cardiomyopathies [194]. In addition, it has been reported that the suppression of Cx43 decreases the activation of the inflammasome, which was preceded by a decrease in the production of ROS and NOX2 [195]. In turn, OS alters the subcellular localization of Cx43 in the heart [196] and modifies the normal transport of Cxs from intercalated discs towards the lateral membrane, affecting excitability of the cardiomyocyte membrane [196]. However, the subcellular location of Cxs has not yet been studied in pathologies that affect skeletal muscle. On the other hand, NOX inhibition in dystrophic cardiomyocytes reduced Cx43 hemichannel activity, probably due to decreased nitrosylation [197].

In dysferlinopathy, a de novo expression of Cxs 39, 43 and 45 has also been reported [151,198,199]. As discussed above, it is accompanied with inflammation [35] and intracellular Ca^2+^ deregulation [151,152]. In human myoblasts lacking dysferlin, the de novo expression of Cx40.1, an ortholog of the rodent Cx39, has been reported in addition to Cxs 43 and 45 [152]. Similarly, skeletal muscles of blAJ mice exhibit an elevated expression of Cx39, 43 and 45 associated to elevated basal intracellular Ca^2+^ in myofibers, and muscle atrophy and lipid accumulation [198,199]. Remarkably, the downregulation of Cx43 and Cx45 prevents increases in intracellular Ca^2+^ and normalizes aberrant lipogenic/muscular commitment, eliminating lipid accumulation and recovering the muscular performance in blAJ mice [199]. These findings suggest that Cx and Panx hemichannels constitute a potential therapeutic target for the treatment of dysferlinopathies. In this regard, the use of boldine, a Cx/Panx-hemichannel blocker that does not affect gap junction channels [200] prevents muscle alterations induced by mutations in the DYSF gene [198].

There are other MDs in which the involvement of Cx and Panx hemichannels has not yet been explored. However, as described above, most MDs share non-specific pathological mechanisms including OS and chronic inflammation in which Cx and Panx hemichannels could participate. In support of this idea the use of boldine has been shown to reduce the ROS levels and inflammation signs in different pathologies [201,202].

## 6. Pharmacological Therapies for MDs: Fight OS, Inflammation and Hemichannels Overexpression as a Potential Alternative

Innovative therapeutic approaches, such as CRISPR/Cas systems and nanomedicine for drug repurposing, are in development for the treatment of MDs [203,204]. However, they are not available in the near future and may not be affordable for many patients. “Nutraceuticals” or functional foods have emerged in the last few years as an alternative to traditional medicine. These can be defined as a food or part of a food that provides health or medical benefits. A large number of nutraceuticals are thought to have anti-inflammatory and/or antioxidant effects [205]. Among them are resveratrol, coenzyme Q10 (CoQ10) and curcumin.

As chronic inflammation, OS and connexin de novo expression are pathological mechanisms that coexist in MDs, an effective therapy should consider the targeting of these three processes. In this regard, an interesting compound is resveratrol, a polyphenol that exhibits both antioxidative and anti-inflammatory properties, as well as inhibits Cx43 hemichannels [191]. This polyphenol reportedly regulates the expression of different types of antioxidative and anti-inflammatory signaling proteins by inhibiting the NF-KB pathway [206,207]. It also activates sirtuin 1 [208], a NAD-dependent deacetylase that regulates transcription factors involved in muscle development, mass and metabolism [209]. In the skeletal muscle, resveratrol regulates the expression of genes involved in mitochondrial biogenesis [210,211], increasing aerobic capacity in mice [210]. Indeed, administration of 0.4 g resveratrol/kg to mdx mice prevent mitochondria accumulation and reduced ROS levels [212,213]. This polyphenol further promotes skeletal muscle hypertrophy in wild-type mice [214] and attenuates, either by itself or in combination with exercise training, the skeletal muscle atrophy induced in different disease animal models [215,216,217,218], including DMD [219]. In mdx mice, it also reduces skeletal muscle necrosis and the expression of inflammatory markers [214]. As evaluated in a pilot randomized controlled trial in senior adults, it enhances the aerobic capacity and improves skeletal muscle mitochondrial function in combination with exercise [220]. Recently, an open-label, single-arm, phase-2 trial was performed with 11 patients with Duchenne, Becker, or Fukuyama MDs who received 500 mg/day of resveratrol, a dose that was increased every 8 weeks to 1000 and 1500 mg/day. After 24 weeks of treatment, motor function, muscular strength and creatine kinase levels significantly improved [221].

It is noteworthy that most of the studies in mdx mice showing that resveratrol improved the dystrophic pathology used doses ranged from 100–400 mg/Kg/day [216,217,218,219], which are 7 to 28-fold higher than maximal doses used in human [220,222]. Only a recent study showed that a low dose of resveratrol (5 mg/Kg/day) reduced exercise-induced skeletal muscle necrosis in mdx mice, as measured by inflammatory infiltrate, myofibres with fragmented sarcoplasm and areas of regenerating myofibres [214]. However, this low dose of resveratrol did not reduce necrosis in the quadriceps of sedentary mdx mice, nor did it increase skeletal muscle hypertrophy, as observed in wild-type mice. Serum creatin kinase activity was not reduced with low resveratrol doses in either sedentary or exercise mdx mice [214]. The latter authors proposed that low doses of resveratrol control exercise-induced inflammation by a signaling pathway different from that of sirtuin 1 [214]. Other studies show that resveratrol displays biphasic dose-dependent effects, exerting antioxidant properties at low concentrations, and increasing oxidative stress at high concentrations [223]. For instance, a low dose of resveratrol (0.07 mg/kg/day) inhibits adenoma development in mice more potently than a dose 200-fold higher (14 mg/kg/day) [224]. Therefore, it is necessary to determine how different doses of resveratrol can impact skeletal muscle function.

Another interesting compound is CoQ10. This is an electron carrier in mitochondrial electron-transport chain [225] and an efficient liposoluble antioxidant [226]. These two CoQ10 properties are beneficial for mitochondrial bioenergetics [227]. Meta-analysis of clinical trials indicates that CoQ10 supplementation increases the levels of total antioxidant capacity and antioxidant defense system enzymes [228,229]. In vitro studies in primary skeletal muscle cell cultures from mdx mice showed that CoQ10 reduced the OS and restored Ca^2+^ levels [230]. In a pilot trial in DMD patients treated with prednisone, CoQ10 showed to increase muscle strength [231]. Therefore, it could be another antioxidant agent that deserves further study in MDs. Furthermore, considering that it exhibits different properties than those described for resveratrol, the effectiveness of these two antioxidant agents could be potentiated when used in combination.

Curcumin is a chemical compound belonging to the group of curcuminoids, which are phenolic pigments produced by plants of the *Curcuma longa* species [232]. Curcumin presents anti-inflammatory and antioxidative properties [233], which seem to rely on its capability to inhibit the activation of NF-κB [234]. Combined with resveratrol, curcumin increases skeletal muscle mass in patients with chronic kidney disease [222]. Furthermore, curcumin has also been shown to significantly reduce the expression of Cx43 by promoting its degradation [197]. Curcumin has been studied in the context of DMD. In 2006 Durham and collaborators fed mdx mice with a 1% (*w*/*w*)-curcumin-supplemented diet, showing an improvement in the muscle contractile properties (compared to mice fed with a control standard chow diet) [235]. Two years later, Pan et al. (2008) reported that the intraperitoneal injection of a higher dose of curcumin (1 mg/Kg) improved specific muscle strength as well as managed to suppress NFκB activation and to reduce the serum TNF-α and IL-1β levels in mdx mice [236]. These data suggest that supplementing the diet with curcumin could help to mitigate the inflammatory and dystrophic signs in DMD. As other neutraceuticals, including resveratrol, curcumin exhibits a very low bioavailability due to its fast metabolization in intestine and liver [237]. However, new delivery systems that improved drug solubility in oral formulations are in development [237].

## 7. Concluding Remarks

MDs are a group of diseases that initially affect mainly skeletal muscles and that are caused by mutations in genes encoding proteins essential for muscle structure and function. The absence or dysfunction of these proteins disturb the muscle integrity and/or homeostasis, provoking a cascade of events including inflammation, overexpression of non-selective channels such as Cx hemichannels, disruption of the ionic asymmetry across the sarcolemma, intracellular Ca^2+^ mishandling and activation of ROS-generating metabolic pathways. Among these latter is mitochondrial dysfunction, which importantly contributes to ROS generation, cell death and consequent muscle degeneration. The type of MD and the MD-causing mutation seem to be critical aspects defining which of these pathways happen first. However, the interaction between them might constitute a positive feedback loop that worsen the disease; therefore, the more advanced is the pathological state the more difficult seems to find a specific molecular target that could abrogate the progression of the disease. Up to now there is no cure for these ailments, yet gene therapy strategies that are in development will probably be accessible in the future. Therefore, the use of existing pharmacological therapies that can disrupt the interplay between these different cellular responses might help to slow down the progression of the muscle dystrophy. In this regard, nutraceuticals that interfere with the signaling pathways inducing OS, inflammation, mitochondria dysfunction or hemichannel activity might mitigate the progress of these diseases and improve the life quality of the patients. Among them, resveratrol, a polyphenol that inhibits the NF-KB pathway and Cx hemichannels, and displays antioxidant and anti-inflammatory properties has shown promising results in clinical trials. However, additional studies are required to determine its optimal dose and treatment schedule.

## Figures and Tables

**Figure 1 biomedicines-10-00507-f001:**
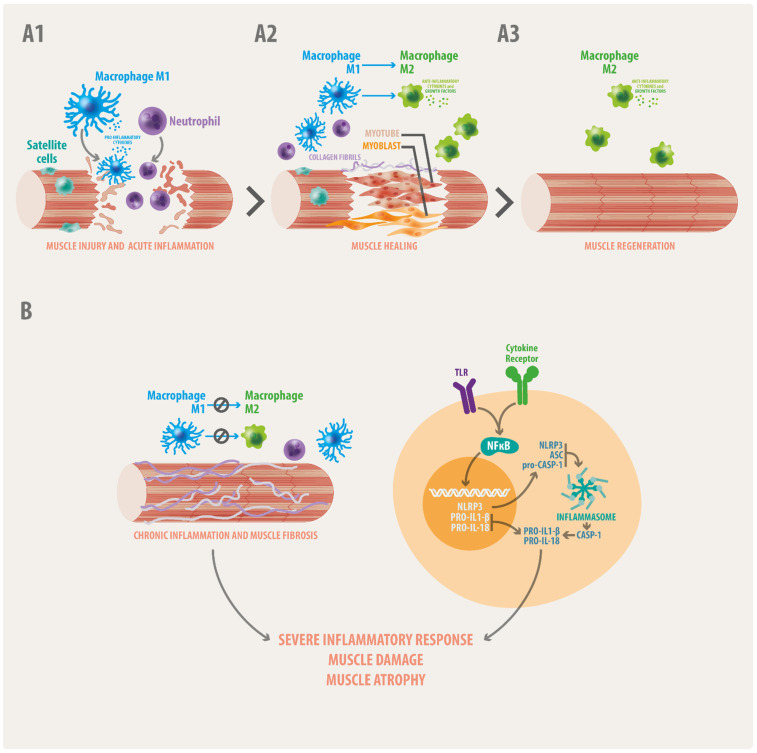
Inflammation in skeletal muscle healing and degeneration. (**A1**–**A3**) Upon muscle injury immune cells infiltrate, monocytes become macrophages with a proinflammatory M1-phenotype that release proinflammatory cytokines and growth factors promoting satellite cell proliferation (**A1**), myotube formation and muscle healing (**A2**) The transition from M1 to M2 pro-resolutive macrophages favors muscle regeneration (**A3**). (**B**) When these processes are deregulated the M1 to M2 transition is suppressed and inflammation becomes chronic, producing accumulation of fibrotic tissue and muscle dysfunction and atrophy. Proinflammatory cytokines bind to their receptors in inflammatory cells as well as in muscle cells, promoting the activation of the NFKB signaling, assembly of the NLRP3 inflammasome, activation of caspase-1 and cleavage of the immature forms of the IL1 family (pro-IL1), thus enhancing inflammation and potentially contributing to muscle damage. Damage signals also activate toll-like receptors (TLRs), promoting the same mechanism.

**Figure 2 biomedicines-10-00507-f002:**
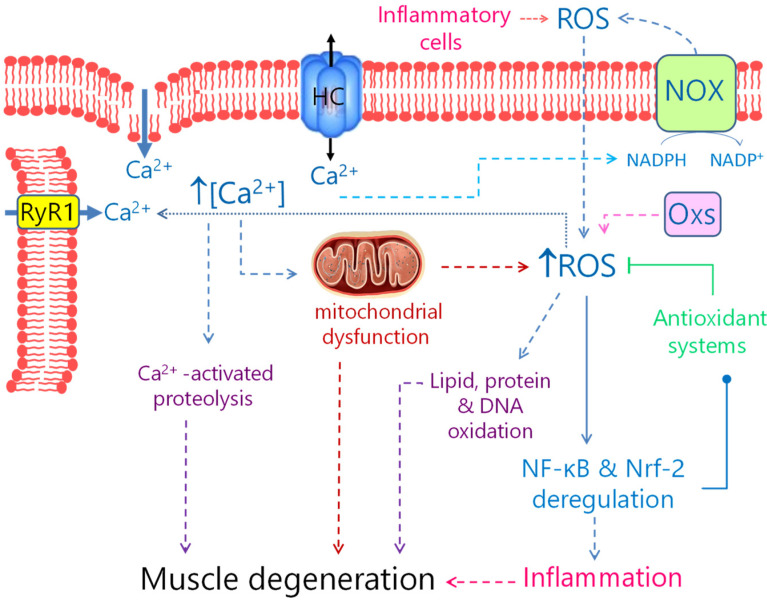
Mechanisms contributing to muscle dystrophy progression. Mutations in proteins that critically regulate skeletal muscle integrity and homeostasis can cause myofiber damage with consecutive accumulation of inflammatory cells that produce ROS. In muscular dystrophies (MDs), such as Duchenne muscular dystrophy (DMD), myofibers also display sarcolemma microtears that allow Ca^2+^ entry. The de novo expression of non-selective channels, such as connexin hemichannels, can further contributes to excessive high cytosolic Ca^2+^ concentrations, which in turn might lead to activation of Ca^2+^-activated proteases, mitochondria dysfunction and reactive oxygen species (ROS) generation from mitochondria, NADPH-oxidase (NOX) and oxidases (Oxs) such as xanthin oxidase. ROS overproduction leads to Ca^2+^ leak from the sarcoplasmic reticulum via ryanodine receptors (RyR1), oxidation of lipids, proteins and DNA, and deregulation of the nuclear factor kappa B (NF-κB) and nuclear factor erythroid-derived 2-related factor 2 (Nrf2) signaling pathways, which regulate the expression of inflammatory mediators and antioxidant enzymes. All these elements contribute to muscle degeneration.

**Table 1 biomedicines-10-00507-t001:** Muscular dystrophy types, genes involved and reported oxidative stress signs, inflammation markers and mitochondria traits in patients’ biopsies and animal models.

Muscular Dystrophy Type	Gene/Protein Associated	Oxidative Stress Signs, Inflammation Markers or Mitochondria Dysfunction
Becker muscular dystrophy (BMD)	DMD/dystrophin	Small inflammatory regions in patients’ muscles [7]. Presence of inflammatory miRNAs [8].
Congenital muscular dystrophy (CMD)	CHKB/choline kinase COL6A1/collagen type VI, subunit α1 COL6A2/collagen type VI, subunit α2 COL6A3/collagen type VI, subunit α3 DPM2/dolichyl-phosphate mannosyltransferase polypeptide 2 DPM3/dolichyl-phosphate mannosyltransferase polypeptide 3 FCMD/fukutin FKRP/fukutin-related protein TGA7/integrin α7 TGA9/integrin α9 LAMA2/laminin α2 chain of merosin LARGE/like-glycosyl transferase PABPN1/polyadenylate binding protein nuclear 1 PTRF/polymerase I and transcript release factor (cavin-1) POMT1/protein-1-O-mannosyl-transferase 1 POMT2/protein-1-O-mannosyl-transferase 2 POMGNT1/protein-O-linked mannose β 1,2-N-aminyltransferase 1 SEPN1/selenoprotein N1	Inflammatory infiltrates in LAMA2-related CMD [9]. Mitochondria dysfunction in LAMA2-related, Megaconial and Ullrich CMDs [10,11,12,13,14].
Distal muscular dystrophies (DiMD)	DYSF/dysferlin GNE/bifunctional UDP-N-acetylglucosamine 2-epimerase/N-acetylmannosamine kinase LDB3/Z-band alternatively spliced PDZ-motif (ZASP) MYH7/myosin heavy chain β TIA1/Tia1 cytotoxic granule-associated rna binding protein TTN/titin	Inflammatory infiltrates in Miyoshi myopathy [15,16].
Duchenne muscular dystrophy (DMD)	DMD/dystrophin	Nucleotide oxidative products, oxidized glutathione and lipid peroxidation [17,18,19]. Overexpression of pro-inflammatory cytokines [20]. Infiltrating inflammatory cells in muscle biopsies of DMD patients [21] Mitochondria abnormality in patients’ biopsies [22]. Mitochondrial dysfunction in mdx mice [23].
Emery-Dreifuss muscular dystrophy (EDMD)	EMD/emerin FHL1/four and a half LIM domain 1 LMNA/lamin A/C SYNE1/nesprin-1 SYNE2/nesprin-2	Altered oxidant status [24,25].
Facioscapulohumeral muscular dystrophy (FSHD)	Unknown/DUX4 Unknown/SMCHD1	Lipid peroxidation, protein carbonylation and DNA oxidation [26]. Mitochondrial dysfunction [26].
Limb-girdle muscular dystrophy (LGMD)	ANO5/anoctamin 5 CAPN3/calpain-3 CAV3/Caveolin-3 DAG1/dystrophin-associated glycoprotein 1 DES/desmin DYSF/dysferlin FKRP/fukutin-related protein FKTN/fukutin LMNA/lamin A/C MYOT/myotilin PLEC1/plectin 1 POMGNT1/protein-O-linked mannose β 1,2-N-aminyltransferase 1 POMT1/protein-1-O-mannosyl-transferase 1 POMT2/protein-O-mannosyl-transferase 2 SGCA/α-sarcoglycan SGCB/β-sarcoglycan SGCD/δ-sarcoglycan SGCG/γ-sarcoglycan TCAP/titin cap TRIM32/tripartite motif-containing 32 TTN/titin	Protein oxidation, lipid peroxidation, altered reduced glutathione and antioxidant enzyme activity in dysferlinopathy patients [27,28,29]. High levels of ROS, protein oxidation, lipid peroxidation, and antioxidant enzyme activity in dysferlin-deficient [30,31,32,33] and calpain-3 deficient mice [34]. Activation of nuclear factor kappa B and inflammasome in dysferin-deficient muscles [35]. Presence of the inflammatory markers Cd68 and Lgals3 in muscles of α- and δ-sarcoglycan-deficient mice [36]. Mitochondria abnormality in skeletal muscle of d.ysferlinopathy patients [37] and calpain-3 deficient mice [34].
Myotonic dystrophy (MiD)	DMPK/myotonin-protein kinase CNBP/cellular nucleic acid-binding protein	Antioxidant imbalance in MiD patients [38]. Mitochondria dysfunction in MiD patients [39].
Occulopharyngeal muscular dystrophy (OMD)	PABPN1/polyadenylate-binding nuclear protein 1	Mitochondria dysfunction in an ice model of OMD [40].

## Data Availability

No applicable.

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
