# Peer review of "Oxidative Stress, Inflammation and Connexin Hemichannels in Muscular Dystrophies"

_biomedicines, 2022, doi:10.3390/biomedicines10020507_

Round 1

Reviewer 1 Report

The authors present a thorough review of the topic, as described in the title.

However, I have some minor comments:

  • Abstract 2nd sentence: Maybe it should be: "MDs can also affect, cardiac, skeletal and respiratory muscles, impairing life-expectancy."
  • In the introduction nearly all closing brackets are missing, i.e. (BMD, ... (CMD, ..., etc. Please check throughout the whole manuscript.
  • In chapter 2, Figure 1A is cited 3 times, and once Figure 1 (only). The 3rd citation. However, the 3rd part of Fig.1A does not show muscle regeneration, but chronic inflammation and muscle fibrosis. The 4th citation (Figure 1) discribes the 3rd part of Fig.1A. Probably the junction between regeneration (in healthy conditions) and muscle dysfunction with chronic inflamation in MDs could be added, and the numbering adjusted, either A - E, or A1-A4 and B.
  • Figure 2 is not referenced in the text.
  • The legende of Figure 2 should mention all abbreviations that are included in the figure. The authors should consider that a review is often read not only by experts in the field, but also by scientist that are not (yet) so familiar with the topic.
  • Chapter 7, 1st sentence: The wording "MDs ... are produced by mutations ... " is probably inappropriate,. Probably "MDs ... are caused by mutations ..." would be better.
  • Chapter 7, 3rd sentence: "Among these later is mitochondrial dysfuntion, ..." Probably the authors are referencing to the sentence before and want to say: "Among these the latter is ..."

Author Response

Reviewer 1:

The authors present a thorough review of the topic, as described in the title.

Response: We acknowledge the reviewer´s comments that have allows to improve our manuscript.

However, I have some minor comments:

  • Abstract 2nd sentence: Maybe it should be: "MDs can also affect, cardiac, skeletal and respiratory muscles, impairing life-expectancy."

Response: We have changed the second sentence in the abstract in order to make it more precisely.

  • In the introduction nearly all closing brackets are missing, i.e. (BMD, ... (CMD, ..., etc. Please check throughout the whole manuscript.

Response: All missing closing brackets are now included.

  • In chapter 2, Figure 1A is cited 3 times, and once Figure 1 (only). The 3rd citation. However, the 3rd part of Fig.1A does not show muscle regeneration, but chronic inflammation and muscle fibrosis. The 4th citation (Figure 1) describes the 3rd part of Fig.1A. Probably the junction between regeneration (in healthy conditions) and muscle dysfunction with chronic inflammation in MDs could be added, and the numbering adjusted, either A - E, or A1-A4 and B.

Response: We appreciate the reviewer's comments in this regard. As she/he suggested the current Figure 1A shows in three stages (A1-A3) the healthy skeletal muscle regeneration in which:

-Fig. 1 A1 shows the acute local inflammation that favors satellite cell proliferation

-Fig. 1 A2 shows the transition of M1 to M2 macrophages that promotes the resolution of inflammation and favors the differentiation of myoblasts to myotubes leading to muscle healing

-Fig. 1 A3 shows how these processes lead to muscle regeneration.

-The current figure 1B shows what happens in the dystrophic context, in which the persistent presence of muscle damage favors the inflammatory processes mediated by the NFkB signaling and the inflammasome, becoming a chronic inflammation chronic that impairs the regeneration and contributes to the muscle dysfunction.

  • Figure 2 is not referenced in the text.

Response: Now Figure 2 is referenced twelfth times along the text.

  • The legend of Figure 2 should mention all abbreviations that are included in the figure. The authors should consider that a review is often read not only by experts in the field, but also by scientist that are not (yet) so familiar with the topic.

Response: Abbreviations are now included in the Figure 2 legend.

  • Chapter 7, 1st sentence: The wording "MDs ... are produced by mutations ... " is probably inappropriate. Probably "MDs ... are caused by mutations ..." would be better.

Response: We have corrected the first sentence in the chapter 7 according to the reviewer´s suggestion.

  • Chapter 7, 3rd sentence: "Among these later is mitochondrial dysfuntion, ..." Probably the authors are referencing to the sentence before and want to say: "Among these the latter is ..."

Response: We have corrected the third sentence in the chapter 7 according to the reviewer´s suggestion.

Reviewer 2 Report

This review of the literature entitled "Oxidative stress, inflammation and connexin hemichannels in muscular dystrophies" highlights a careful research on the topic.
The authors have performed a careful and complete systematic search of the literature which determines an important enrichment  to the readers.

Improve, in general, native English. 

Author Response

Reviewer 2:

This review of the literature entitled "Oxidative stress, inflammation and connexin hemichannels in muscular dystrophies" highlights a careful research on the topic.
The authors have performed a careful and complete systematic search of the literature which determines an important enrichment to the readers.

Improve, in general, native English. 

Response: We appreciate the positive comments of this reviewer. We have corrected the English grammatical errors.